# A Comparison of Electronic and Traditional Stethoscopes in the Heart Auscultation of Obese Patients

**DOI:** 10.3390/medicina55040094

**Published:** 2019-04-05

**Authors:** Eglė Kalinauskienė, Haroldas Razvadauskas, Dan J. Morse, Gail E. Maxey, Albinas Naudžiūnas

**Affiliations:** 1Department of Internal Medicine, Lithuanian University of Health Sciences, LT-47144 Kaunas, Lithuania; haroldas.raz@gmail.com (H.R.); albinas.naudziunas@lsmuni.lt (A.N.); 23M Center, St. Paul, MN 55144, USA; danjmorse22@mmm.com (D.J.M.); gemaxey@mmm.com (G.E.M.)

**Keywords:** auscultation, electronic stethoscope, acoustic stethoscope, heart murmurs, obese patients

## Abstract

*Background and objectives:* As the prevalence of obesity is increasing in a population, diagnostics becomes more problematic. Our aim was to compare the 3M Littmann 3200 Electronic Stethoscope and 3M Littman Cardiology III Mechanical Stethoscope in the auscultation of obese patients. *Methods*. A total of 30 patients with body mass index >30 kg/m^2^ were auscultated by a cardiologist and a resident physician: 15 patients by one cardiologist and one resident and 15 patients by another cardiologist and resident using both stethoscopes. In total, 960 auscultation data points were verified by an echocardiogram. Sensitivity and specificity data were calculated. *Results.* Sensitivity for regurgitation with valves combined was higher when the electronic stethoscope was used by the cardiologist (60.0% vs. 40.9%, *p* = 0.0002) and the resident physician (62.1% vs. 51.5%, *p* = 0.016); this was also the same when stenoses were added (59.4% vs. 40.6%, *p* = 0.0002, and 60.9% vs. 50.7%, *p* = 0.016, respectively). For any lesion, there were no significant differences in specificity between the electronic and acoustic stethoscopes for the cardiologist (92.4% vs. 94.2%) and the resident physician (93.6% vs. 94.7%). The detailed analysis by valve showed one significant difference in regurgitation at the mitral valve for the cardiologist (80.0% vs. 56.0%, *p* = 0.031). No significant difference in specificity between the stethoscopes was found when all lesions, valves and both physicians were combined (93.0% vs. 94.4%, *p* = 0.30), but the electronic stethoscope had higher sensitivity than the acoustic (60.1% vs. 45.7%, *p* < 0.0001). The analysis when severity of the abnormality was considered confirmed these results. *Conclusions.* There is an indication of increased sensitivity using the electronic stethoscope. Specificity was high using the electronic and acoustic stethoscope.

## 1. Introduction

With advances in technology, the electronic stethoscope models of a new generation were created with the aim to avoid the previous auditory limitations of traditional stethoscopes and to apply the possibilities of new electronic technologies. The first electronic stethoscopes already had the ability to filter sound frequencies, eliminate background noise and amplify heart sounds—something impossible with acoustic stethoscopes. The new 3M Littmann 3200 electronic stethoscope also has the ability to transmit heart sounds via Bluetooth and this has been approved by the Food and Drug Administration (FDA) in the USA. Newer models of electronic stethoscopes have the ability to store and replay heart sounds away from the patient via an external source or computer. This can help in the teaching of medical students by the bedside [1,2]. The idea of combining such a stethoscope with computer software that could visualize the murmur and heart sounds was proposed already several years ago for undergraduate teaching [2].

Studies comparing acoustic and electronic stethoscopes in clinical practice are scarce. The study by Grenier et al. [3] compared electronic stethoscopes with standard devices and concluded that acoustic stethoscopes were preferred. However, they proposed that an ideal stethoscope would be a combination of the advantages of both acoustic and electronic stethoscopes. Another two studies [4,5] found no difference in diagnostic accuracy, but these studies were not performed on obese patients. The sensitivity and specificity of auscultation depends on the expertise of the examiner [6], but obesity also reduces the transmission of sound [7]. It is clear that the auscultation of obese patients is more problematic than the auscultation of the non-obese [7]. In order to improve the auscultation of obese patients, one possibility could be the use of new electronic technologies. We failed to find any study investigating the auscultation of obese patients using these two different types of stethoscopes for heart murmurs. However, the population’s body mass index (BMI) is increasing in an ever-greater number of countries [8,9,10]. Therefore, our main aim was to compare the ability of auscultating heart murmurs in obese subjects using both electronic and acoustic stethoscopes. We also compared the benefit of using two different types of stethoscope for an experienced cardiologist in contrast to a less experienced but younger resident physician. This is important because with age, we have a gradual decline in our hearing ability [11,12].

## 2. Methods

### 2.1. Study Population and Design

Hypothesis generating pilot study (ClinicalTrials.gov Identifier NCT01665820) aimed to investigate the effect of using an electronic against an acoustic stethoscope in the ability of physicians to hear heart murmurs in obese patients. The consecutive patients arriving at Kaunas Clinical Hospital were considered for inclusion into this study if they had a BMI greater than 30 kg/m^2^ (obese), were older than 18 years, were referred for an echocardiogram and agreed to participate in the study. Patients gave written informed consent for prospective inclusion in the study. The exclusion criteria were as follows: if the investigator believed the subject should not be included (a severe status of patient) or was unsuitable for inclusion (echo-negative patient). The study was approved by Kaunas Regional Biomedical Research Ethics Committee (approval No. BE-2-91).

Each subject received 4 auscultation examinations (Figure 1). Two auscultations were done by a cardiologist with about 20 years of experience (one—21, another—19 years of experience) using both an acoustic traditional stethoscope (3M Littman Cardiology III Mechanical Stethoscope, 3M Health Care, St. Paul, MN, USA) and an electronic stethoscope (3M Littmann 3200 Electronic Stethoscope, 3M Health Care, St. Paul, MN, USA). Two additional auscultations were done by a 3rd-year medical resident doctor also using both an acoustic traditional stethoscope and an electronic stethoscope. Half of the patients were auscultated by one cardiologist and one resident, and half, by another cardiologist and another resident. Based on the randomization for each subject, the auscultation may begin with either the 3M Littmann 3200 Electronic or 3M Littmann Cardiology III Mechanical stethoscope. Physicians had a 2-week period to gain experience in using the electronic stethoscope before the commencement of the study. Each auscultation consisted of heart murmurs being listened to in the following sites: mitral (apex), aortic (right second intercostal space), pulmonary (left second intercostal space), tricuspid (lower left sternal border), and Erb’s (left third intercostal space).

All physicians used the same ordinary methodology of Lithuanian University of Health Sciences: (1) Using the bell listen first to the apex (mitral area) just above the apex beat (palpate the apex beat), then at the second right interspace parasternally (aortic area), at the second left interspace parasternally (pulmonic area), at the third interspace adjacent to the left sternal border (Erb‘s area), and, finally, at the left parasternal area at the lower part of the sternum (tricuspid area) in the supine position. (2) Shift to the diaphragm and return to all these areas. (3) Ask the patient to exhale completely and stop breathing, listen at the apex and aortic areas, and Erb‘s area with the bell and the diaphragm; ask the patient to inhale completely and stop breathing, listen at the pulmonic and tricuspid areas with the bell and the diaphragm. (4) Ask the patient to roll partly onto the left side, listen at the apex with the bell and the diaphragm, then also ask the patient to exhale completely and stop breathing, and again listen with the bell and the diaphragm. (5) Ask the patient to sit up, lean forward and put his/her arms on the head, listen at the aortic and Erb‘s areas only with the diaphragm, also repeat the auscultation at these areas in full held expiration. 

Physical conditions for all listeners were as in a real-life: all auscultations were performed in an ordinary room of patients. Physicians were allowed to listen at a particular site more times until they were happy with the sound quality during one auscultation. There was a time-frame of 3 h or less between the completion of the first and second auscultation so as to minimize bias and to keep the results as accurate as possible. After completion of all auscultations, the subject underwent echocardiography. Echocardiography was done by a different clinical team that had no access to the information obtained from the auscultations completed for the study. 

In clinical practice, echocardiography is the standard for establishing the cause and severity of a murmur. The 2006 American College of Cardiology/American Heart Association guidelines on the management of patients with valvular heart disease (with 2008 focused update) included recommendations for the use of echocardiography in patients with symptomatic and asymptomatic murmurs [13]. Thus, the auscultation data we received were verified using an echocardiogram.

### 2.2. Statistical Analysis

The echocardiography clinical report form allowed for the reporting of three lesions. When comparing the auscultation results to the echocardiography findings, each echocardiography lesion found was included in the analysis. Thus it was possible for a physician to correctly identify one abnormality but not another. Each valve was analyzed separately.

Sensitivity/specificity analysis was conducted in two ways: (1) Was there an abnormality observed/heard or not? (2) Was there a correct identification of appropriate severity?

The first analysis aimed to observe if there was an abnormality observed/heard—the severity of the abnormality was not considered. Tables were produced by valve for both regurgitation and stenosis, by abnormality with all four valves combined and with all four valves and regurgitation and stenosis combined. McNemar’s test was used to compare the stethoscopes for sensitivity and specificity for all combinations.

In the second analysis, correct identification of appropriate severity was considered. For this analysis, Table 1 shows the definition for correct identification of an echocardiography finding by auscultation. As there were only a few cases of stenosis observed, this was done for regurgitation only. As with the first analysis, the data were grouped in several ways. 

The Erb’s site was also used in this study for auscultation. If a diastolic regurgitation was heard at the Erb’s site, this was called an aortic regurgitation and the severity of the regurgitation was given the larger of the Erb’s value and the aortic location value. If a systolic regurgitation was heard at the Erb’s site, this was called a mitral regurgitation and the severity of the regurgitation was given the larger of the Erb’s value and the mitral location value.

The chi-squared test was used to compare stethoscopes where the data was down to a 2 × 3 table. The significance level was at <0.05.

## 3. Results

Thirty patients, 20 men and 10 women, were included in the study (Table 2). No one patient was excluded from the study. The mean age of the subjects was 68.27 years and their mean BMI was 34.53 kg/m^2^. The most common reasons for hospitalization were as follows: dyspnea, chest pain, and abnormal electrocardiogram. Echocardiography findings showed that the most frequent valve abnormalities were mitral regurgitation, tricuspid regurgitation, and aortic regurgitation.

The total number of discrete heart auscultation data points reached 960—this originated from 4 different auscultations done on 30 patients, each having auscultations of 4 valves, with 2 abnormalities being regurgitation and stenosis. The echocardiographic data confirmed 189 counts of abnormalities being heard and 771 abnormalities not being heard on auscultation in the cases of 276 lesions observed and 684 lesions not observed during echocardiography in the first analysis. 

In the detailed sensitivity and specificity analysis to identify the existence of regurgitation by valve, the only *p* value less than 0.05 was for sensitivity for regurgitation at the mitral valve for the cardiologist (*p* = 0.031, Table 3). Six regurgitations were correctly heard with the electronic stethoscope, but missed with the acoustic stethoscope.

Only 3 subjects had stenosis and they were all aortic. There were no other stenoses observed using echocardiography. No significant differences in sensitivity or specificity in the identification of stenosis by valve for both physicians when comparing acoustic and electronic stethoscopes were documented (Table 4).

Sensitivity for regurgitation when all valves were combined was higher with the electronic stethoscope for the cardiologist (60.6% vs. 40.9%, *p* = 0.0002) and for the resident (62.1% vs. 51.5%, *p* = 0.016). Thirteen regurgitations were correctly heard by the cardiologist with the electronic stethoscope, but missed with the acoustic stethoscope and seven regurgitations were correctly heard by the resident with the electronic stethoscope, but missed with the acoustic stethoscope. When stenoses were added sensitivity was higher also with the electronic stethoscope for the cardiologist (59.4% vs. 40.6%, *p* = 0.0002) and the resident (60.9 vs. 50.7%, *p* = 0.016).

There were no significant differences in specificity for regurgitation with valves combined between the acoustic and electronic stethoscopes for the cardiologist (90.7% vs. 81.5%) and for the resident (92.6% vs. 87.0%). In addition, there were no significant differences in specificity for stenosis with valves combined between the acoustic and electronic stethoscopes for the cardiologist (95.7% vs. 97.4%) and for the resident (95.7% vs. 96.6%). For any cardiac abnormality, there were no significant differences in specificity between the acoustic and electronic stethoscopes for the cardiologist (94.2% vs. 92.4%) and for the resident (94.7% vs. 93.6%).

Sensitivity for regurgitation when all valves and both physicians were combined was higher with the electronic stethoscope (61.4% vs. 46.2%, *p* < 0.0001). Twenty regurgitations were correctly heard with the electronic stethoscope but missed with the acoustic stethoscope. Sensitivity for stenosis was the same for both stethoscopes (33.3% vs. 33.3%), but there were only six cases of auscultation of echo-positive stenosis. 

Thus, for any cardiac abnormality, the electronic stethoscope had higher sensitivity (60.1% vs. 45.7%, *p* < 0.0001) as compared to the acoustic stethoscope when all valves and both physicians were combined. Specificity for regurgitation when all valves and both physicians were combined was higher for the acoustic than the electronic stethoscope (91.7% vs. 84.3%, *p* = 0.008). Eight regurgitations were incorrectly heard with the electronic stethoscope, but correctly not heard with the acoustic stethoscope. There was no significant difference in specificity for stenosis between the acoustic and electronic stethoscopes when all valves and both physicians were combined (95.7% vs. 97.0%). When all lesions and all valves and both physicians were combined, no significant difference in specificity between the acoustic and electronic stethoscopes was observed (94.4% vs. 93.0%, *p* = 0.30).

The analysis when severity of the abnormality was considered confirmed these results. The only *p* value less than 0.05 was for the cardiologist (*p* = 0.01) when analyzing regurgitation (Table 5).

When severity and physicians were combined, the only *p* value less than 0.05 was for the regurgitation of the mitral valve (*p* = 0.016) (Table 6).

Correct identification of regurgitation with severity, valve and physicians combined was in 64.58% of the cases with the electronic stethoscope and in 60.42% of the cases with the acoustic stethoscope. There were 22.92% of false-negative cases with the electronic stethoscope and 33.75% with the acoustic stethoscope. However, there were more false-positive cases with the electronic than the acoustic stethoscope: 12.50% vs. 5.83%. The *p* value was 0.004 indicating that the move to the right for the electronic stethoscope compared to the acoustic stethoscope was statistically significant.

No obvious effects on sensitivity and specificity were seen with respect to the use order of the stethoscope in the auscultation of regurgitation. Sensitivity of the acoustic stethoscope was 50.0% when used as the first stethoscope and 40.7% when used as the second stethoscope as compared to the electronic stethoscope which had 55.6% and 65.4% sensitivity, respectively. Specificity of the acoustic stethoscope was 91.4% when used first and 92.0% when used second in comparison to the electronic stethoscope having a specificity of 86.0% and 82.8%, respectively.

There were no significant differences in auscultation sensitivity or specificity between the two cardiologists or the two resident physicians (Figure 2).

## 4. Discussion

Our findings suggest that there is an indication of increased sensitivity using the electronic stethoscope. The electronic stethoscope had higher sensitivity than the acoustic stethoscope for regurgitation with valves combined for both the cardiologist and the resident in the analysis when severity of the abnormality was not considered. The sensitivity comparison when stenoses were added did not change these results since there were no discordant results for echo-positive stenoses. The detailed analysis by valve showed only one significant difference for regurgitation at the mitral valve for the cardiologist but not the resident physician. Mitral regurgitation was the most common valve abnormality in our study and cardiologists were older than 45 years. As we gradually lose our high-frequency hearing with age [11,12], an electronic stethoscope might be especially useful for older physicians, particularly for regurgitations which feature high-frequency murmurs [14]. The analysis when severity of the abnormality was considered confirmed the results seen with the first analysis. As there were only a few cases of stenosis observed, the second analysis was done for regurgitation only.

There were no statistical differences in specificity between the two types of stethoscopes for the cardiologist and the resident physician in both analyses. However, the specificity for regurgitation when all valves and both physicians were combined was higher with the acoustic as compared to the electronic stethoscope in the analysis, when severity of the abnormality was not considered. This could be due to inexperience with the electronic stethoscope. The physicians had only a 2-week period to familiarize themselves with the electronic stethoscope before the beginning of the study. We think that this is an insufficient amount of time for someone to accustom themselves to the electronic stethoscope. Hence, with more experience, there may be increased specificity using the electronic stethoscope.

The results of this study are supported by a small study that concluded that the electronic stethoscope was more sensitive than the traditional one in dogs [15]. However, it was an animal study using a different brand of electronic stethoscopes. Another recent study has concluded that digital recordings made with electronic stethoscopes are less sensitive but comparably specific to the conventional stethoscopes at detecting abnormal heart sounds in cats [16]. Despite a different methodology being employed compared to ours, the same brand of stethoscopes was used. The earlier human studies mentioned in the introduction showed no advantages of an electronic stethoscope over an acoustic stethoscope [3,4,5]. There is a lack of heart auscultation studies with modern stethoscopes in obese patients. One study on obese patients used a different method—audiocardiography [17]. The authors concluded that for senior physicians acoustic cardiography performed with an electronic device was not helpful in assisting the cardiovascular examination of the morbidly obese. However, they used another method and assessed heart sounds (S3, S4), but not murmurs. In our study, we assessed the ability of physicians to hear heart murmurs in obese patients. 

Our study employed echocardiography as the gold standard to verify the results of auscultation [13]. The echocardiography clinical report form allowed for the reporting of three lesions, with the addition of stenosis and regurgitation, 6 echocardiography outcomes were possible. The auscultation clinical report form allowed eight possible auscultation outcomes. In clinical practice, we have 3 degrees (mild, moderate, and severe) of stenosis severity by auscultation as well as by echocardiography. However, in cases of regurgitation we have three degrees by auscultation but four degrees by echocardiography. Therefore, the exact data comparison is difficult. Hence, we created a table to standardize our comparisons between echocardiography and auscultation results (Table 1). It is possible that an easier solution to such comparisons will be found in future studies.

Our experience with this electronic stethoscope showed that despite its many possibilities, the electronic stethoscope is easy and convenient to use. We liked the auscultation with a bell and membrane together as it provided a better assessment of murmurs’ connection with heart sounds and its possibility to increase the sound in obese patients, not to mention its ability to monitor the heart rate during auscultation. This is however impossible to achieve with an acoustic stethoscope.

The capability of newer generation of electronic stethoscope models to store and playback heart sounds away from the patient may be useful for teaching and patient monitoring. These stethoscopes allowed us to document not only murmurs, but as we discovered, sometimes it enabled us to document the elusive arrhythmias. Visualization of the murmurs could be useful for physicians in cases of gradual and continuous loss of hearing with age. The ability to transmit heart murmurs via Bluetooth technology also gives us an opportunity for consultations to occur among physicians over larger distances. 

Recently, the new advantage of electronic stethoscopes over the conventional stethoscope has been highlighted: the possibility to use them for screening obstructive coronary artery disease [18]. Conventional stethoscopes lack the auscultation power to detect intracoronary murmurs of turbulent blood flow occurring due to hemodynamically significant coronary artery disease. The overall sensitivity of 89.5% was shown using the electronic stethoscope to ascertain coronary artery disease of >50% in any major epicardial artery [19].

Summarizing, our study suggests that there is great potential in electronic stethoscopes for heart auscultations. Going forward, we can expect electronic stethoscopes to become more widely used in medicine.

### Limitations

We analyzed 960 auscultations in our study; however, these auscultations were obtained from 30 patients only and we had only 2 cardiologist/resident pairs because it was a pilot study. It is a major limitation of the study. Another limitation is that half of the patients were seen by one cardiologist/resident pair and another half by another cardiologist/resident pair. Furthermore, the use of 8 possible auscultation outcomes and 3 possible echocardiography outcomes inflates the apparent power. Moreover, the physicians had only a 2-week period to familiarize with the electronic stethoscope before the beginning of the study. Therefore, the analyses were exploratory and were only used in the generation of hypotheses for later confirmation.

## 5. Conclusions

There is an indication of increased sensitivity using the electronic stethoscope in obese patients. Specificity was high using the electronic stethoscope as well as using the acoustic. There is also an indication that electronic stethoscopes might be especially useful for older physicians, particularly for regurgitations that are high-frequency murmurs.

This hypothesis should be confirmed in a larger study. Physicians should also be provided with a longer period to familiarize themselves with the electronic stethoscope.

## Figures and Tables

**Figure 1 medicina-55-00094-f001:**
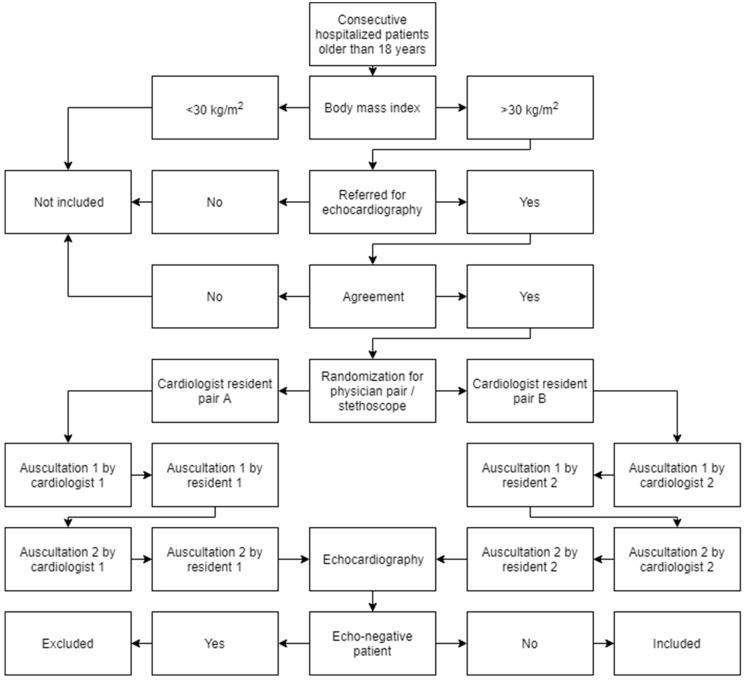
Study flow chart. Patients could be excluded at any step of the study due to a severe status of patient. Based on the randomization for each subject, auscultation may begin with either the 3M Littmann 3200 Electronic or 3M Littmann Cardiology III Mechanical stethoscope (auscultation 1 was performed with one, auscultation 2 was performed with another stethoscope).

**Figure 2 medicina-55-00094-f002:**
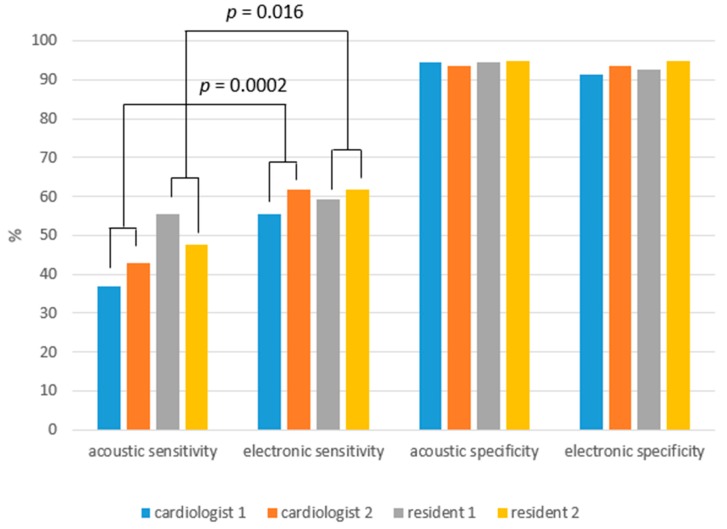
Sensitivity and specificity of heart auscultation using the electronic and acoustic stethoscopes by physicians. Comparison of auscultation data (*n* = 960) with echocardiographic data of 30 patients when regurgitation, stenosis and valves are combined. No *p* < 0.05 comparing the cardiologist 1 with the cardiologist 2, and comparing the resident 1 with the resident 2, and comparing the specificity between the electronic and acoustic stethoscopes for the both cardiologists combined and for the both residents combined.

**Table 1 medicina-55-00094-t001:** The definition for correct identification of an echocardiography finding by auscultation.

Echocardiography: Degree of Regurgitation	Auscultation: Regurgitation Severity			
	Not Observed	Mild	Moderate	Severe
Not Observed	Correct	False Positive	False Positive	False Positive
I Degree	False Negative	Correct	False Positive	False Positive
II Degree	False Negative	Correct	Correct	False Positive
III Degree	False Negative	False Negative	Correct	Correct
IV Degree	False Negative	False Negative	False Negative	Correct

**Table 2 medicina-55-00094-t002:** Baseline characteristics of patients.

Variable	Value
Age, mean (SD) (range), years	68.27 (12.09) (45.00–96.00)
Male, *n* (%)	20 (66.67)
Height, mean (SD) (range), m	1.69 (0.08) (1.47–1.86)
Weight, mean (SD) (range), kg	98.77 (13.66) (71.00–130.00)
Body mass index, mean (SD) (range), kg/m^2^	34.53 (4.02) (30.70–46.60)
Reasons for hospitalization, *n* (%)	
Dyspnea	25 (83.33)
Chest pain	23 (76.67)
Abnormal electrocardiogram	17 (56.67)
Leg edema	11 (36.67)
Fatigue	9 (30.00)
Hypertension	4 (13.33)
Abnormal chest x-ray	3 (10.00)
Syncope	2 (6.67)
Other (by one case)	2 (6.67)
Disease entities, *n* (%)	
Coronary artery disease	26 (86.67)
Primary hypertension	23 (76.67)
Heart failure	17 (56.67)
Diabetes	7 (23.33)
Pneumonia	3 (10.00)
Bronchitis	3 (10.00)
Other (by one case)	2 (6.67)
Echocardiography findings, *n* (%)	
Mitral regurgitation	25 (83.33)
Tricuspid regurgitation	20 (66.67)
Aortic regurgitation	19 (63.33)
Pulmonic regurgitation	2 (6.67)
Aortic stenosis and regurgitation	2 (6.67)
Aortic stenosis	1 (3.33)

SD: standard deviation.

**Table 3 medicina-55-00094-t003:** Sensitivity/Specificity for identification of the existence of regurgitation by valve.

Valve n (echo-positive/echo-negative)	Sensitivity Acoustic vs. Electronic, %		Specificity Acoustic vs. Electronic, %	
	Cardiologist	Resident	Cardiologist	Resident
Mitral 25/5	56.0 vs. 80.0 *	76.0 vs. 84.0	80.0 vs. 20.0	60.0 vs. 40.0
Aortic 19/11	26.3 vs. 47.4	26.3 vs. 36.8	100.0 vs. 100.0	100.0 vs. 100.0
Pulmonary 2/28	0 vs. 0	0 vs. 0	89.3 vs. 85.7	96.4 vs. 92.9
Tricuspid 20/10	40.0 vs. 55.0	50.0 vs. 65.0	90.0 vs. 80.0	90.0 vs. 80.0

* *p* < 0.05.

**Table 4 medicina-55-00094-t004:** Sensitivity/Specificity for identification of the existence of stenosis by valve.

Valve, *n* (echo-positive/echo-negative)	Sensitivity Acoustic vs. Electronic, %		Specificity Acoustic vs. Electronic, %	
	>Cardiologist	>Resident	>Cardiologist	>Resident
Mitral 0/30			93.3 vs. 96.7	96.7 vs. 96.7
Aortic 3/27	33.3 vs. 33.3	33.3 vs. 33.3	92.6 vs. 92.6	88.9 vs. 88.9
Pulmonary 0/30			96.7 vs. 100.0	96.7 vs. 100.0
Tricuspid 0/30			100.0 vs. 100.0	100.0 vs. 100.0

**Table 5 medicina-55-00094-t005:** Identification of regurgitation with severity and valves combined by physician.

	Cardiologist			Resident		
	Auscultation Results, *n* (%)			Auscultation Results, *n* (%)		
	False Negative	Correct	False Positive	False Negative	Correct	False Positive
Acoustic	42 (35.00)	72 (60.00)	6 (5.00)	39 (32.50)	73 (60.83)	8 (6.67)
Electronic	26 (21.67)	77 (64.17)	17 (14.17)	29 (24.17)	78 (65.00)	13 (10.83)

*p* < 0.05 was for the cardiologist.

**Table 6 medicina-55-00094-t006:** Identification of regurgitation with severity and physicians combined by valve.

	Mitral Valve *n* %			Aortic Valve *n* %			Pulmonary Valve *n* %			Tricuspid Valve *n* %		
	Auscultation Results			Auscultation Results			Auscultation Results			Auscultation Results		
	False Negative	Correct	False Positive	False Negative	Correct	False Positive	False Negative	Correct	False Positive	False Negative	Correct	False Positive
Acoustic	20 33.33	35 58.33	5 8.33	28 46.67	31 51.67	1 1.67	4 6.67	52 86.67	4 6.67	29 48.33	27 45.00	4 6.67
Electronic	10 16.67	35 58.33	15 25.00	22 36.67	36 60.00	2 3.33	4 6.67	508 3.33	6 10.00	19 31.67	34 56.67	7 11.67

*p* < 0.05 was only for the mitral valve.

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
