# Peer review of "A Comparison of Electronic and Traditional Stethoscopes in the Heart Auscultation of Obese Patients"

_medicina, 2019, doi:10.3390/medicina55040094_

Round 1
Reviewer 1 Report
My comments have been mainly addressed in the revised manuscript. I have no further comments.
Author Response
Thank you very much.
Reviewer 2 Report
General comments:
This is a study that compared two types of stethoscope, Littman digital and acoustic stethoscope, with the aim of assessing cardiac pathology among obese patients. It is important to acknowledge that this study was sponsored by 3M (stethoscope manufacturer) and two co-authors are employees of the company, which is disclosed in competing interests section and in acknowledgments.
In general, it is a well-written article and balanced, however, in this study, both products that were used were from the same manufacturer.
Ethical/legal disclosures:
Ethical and legal disclosures are appropriate.
Statistical methods:
Statistical methods applied are appropriate.
Specific comments:
- Auscultation is a very subjective skill that is dependent on proficiency of physician (in this case cardiologists and residents), therefore, this might have a great influence on sensitivity, and especially specificity of the findings, however, I do appreciate the fact that both attending cardiologists were well-balanced in terms of experience (21 vs. 19 years of work) while residents were in the same level of training and authors had a decent run-in period to get themselves familiar with the stethoscope
- Why did the authors only limited this study on heart auscultation and not lung auscultation as well?
- One thing from Methods is not clearly described. Authors say that each subject received four auscultation examinations, two from 1 doctor (one with electronic and another with an acoustic stethoscope) and two from a resident (one with electronic and another with the acoustic stethoscope). The authors go on to say that half of the patients were seen by one cardiologist/resident pair and another half by another cardiologist/resident pair.
I am having a problem with such a methodology. It would be the best approach if all patients underwent auscultation by all four auscultation physicians and then the analysis would be more worthwhile and you could then also analyze for inter-observer and intra-observer variability and bias among listeners. In this way, since each pair listen to different/other patients, this analysis is not possible. This should be acknowledged as a shortcoming of the study
- I would advise authors to make a study flow-chart figure in which they should depict all inclusion/exclusion criteria and interventions – what was done to which patient, in which order and how this is unclear and pretty fuzzy from the text
- Authors should describe what was the approximate percentage or specific number (if they have one) on the patients that were excluded from the study (e.g. approached but not satisfying criteria)
- Was this a sample of consecutively enrolled patients?
- Similarly to a comment above, randomization procedure should be detailed and also explained/graphed within the study flow-chart figure
- Were these auscultations undertaken under the same physical conditions for all listeners, e.g. the room, noise, ambient, etc? Also, more information on methodology itself, since that momentum is important in this type of study? Were patients seated, semi-recumbent, upright, all of these details matter and should be defined.
- Were physicians allowed to listen at a particular site more times until they are happy with the sound quality or were they allowed only to listen once at a certain spot? This is important since we should know what was the auscultation baseline in this study
- While I do understand that this is a pilot study, limitation of having only two physicians and residents and only 30 patients (despite generating a lot of data points) is limiting conclusion strength and should be emphasized as a major limitation
- While authors did mention chief reasons of hospitalization of their 30 subjects, mostly being admitted for dyspnea and chest pain, I would advise authors to prepare a Table that would contain all baseline characteristics of variables that were recorded for this patient cohort, as well as disease entities (e.g. heart failure, etc.) – were these patients ambulatory or acute?
- Figure 1. Please change the coloring of the columns and make them more distinct. It is difficult to follow what is going on over there and I think our readers would appreciate the more colorful depiction of this figure since it is very difficult to follow it in this current form.
Author Response
We are very grateful for all the comments given by the reviewer. We hope that all questions are answered properly and believe that with the help of the reviewer our manuscript is improved considerably. Our answers to the remarks by the reviewer are highlighted in red.
Specific comments:
- Auscultation is a very subjective skill that is dependent on proficiency of physician (in this case cardiologists and residents), therefore, this might have a great influence on sensitivity, and especially specificity of the findings, however, I do appreciate the fact that both attending cardiologists were well-balanced in terms of experience (21 vs. 19 years of work) while residents were in the same level of training and authors had a decent run-in period to get themselves familiar with the stethoscope
Yes, we agree that auscultation is a very subjective skill that depends on the proficiency of a physician and might have a great influence on sensitivity and especially specificity of the findings; however, as you noted both attending cardiologists were well-balanced in terms of experience (21 vs. 19 years of work) while residents were at the same level of training and all of them had a 2-week period to get themselves familiar with the stethoscope. We wrote in Conclusions that our hypothesis should be confirmed in a larger scale study. Physicians should also be provided with a longer period to familiarize themselves with the electronic stethoscope.
- Why did the authors only limited this study on heart auscultation and not lung auscultation as well?
We limited this study on heart auscultation, because the first and corresponding author is a cardiologist.
- One thing from Methods is not clearly described. Authors say that each subject received four auscultation examinations, two from 1 doctor (one with electronic and another with an acoustic stethoscope) and two from a resident (one with electronic and another with the acoustic stethoscope). The authors go on to say that half of the patients were seen by one cardiologist/resident pair and another half by another cardiologist/resident pair.
I am having a problem with such a methodology. It would be the best approach if all patients underwent auscultation by all four auscultation physicians and then the analysis would be more worthwhile and you could then also analyze for inter-observer and intra-observer variability and bias among listeners. In this way, since each pair listen to different/other patients, this analysis is not possible. This should be acknowledged as a shortcoming of the study
If all patients had underwent auscultation by all 4 auscultation physicians, it would have resulted in 8 auscultation examinations for each patient instead of 4 auscultation examinations. As it was a pilot study performed in a real-life, we tried to bother our patients as little as possible. There were no significant differences in auscultation sensitivity or specificity between the two cardiologists or the two resident physicians (Figure 2). However, we added to Limitations that auscultations were obtained from 30 patients only and we had only 2 cardiologist/resident pairs because it was a pilot study. It is a major limitation of the study. Another limitation is that half of the patients were seen by one cardiologist/resident pair and another half by another cardiologist/resident pair.
- I would advise authors to make a study flow-chart figure in which they should depict all inclusion/exclusion criteria and interventions – what was done to which patient, in which order and how this is unclear and pretty fuzzy from the text
We added a study flow chart (Figure 1), according to your suggestion.
- Authors should describe what was the approximate percentage or specific number (if they have one) on the patients that were excluded from the study (e.g. approached but not satisfying criteria)
No one patient was excluded from the study. We added this sentence to Results.
- Was this a sample of consecutively enrolled patients?
Yes, this was a sample of consecutively enrolled patients. We updated information on this in Methods: The consecutive patients arriving at Kaunas Clinical Hospital were considered for inclusion into this study if they had a BMI greater than 30 kg/m2 (obese), were older than 18 years, were referred for an echocardiogram and agreed to participate in the study.
- Similarly to a comment above, randomization procedure should be detailed and also explained/graphed within the study flow-chart figure
We added the randomization procedure to the study flow chart (Figure 1).
- Were these auscultations undertaken under the same physical conditions for all listeners, e.g. the room, noise, ambient, etc? Also, more information on methodology itself, since that momentum is important in this type of study? Were patients seated, semi-recumbent, upright, all of these details matter and should be defined.
According to your suggestion, we added a more detailed description of auscultation methodology in Methods: All physicians used the same ordinary methodology of Lithuanian University of Health Sciences: 1) Using the bell listen first to the apex (mitral area) just above the apex beat (palpate the apex beat), then at the second right interspace parasternally (aortic area), at the second left interspace parasternally (pulmonic area), at the third interspace adjacent to the left sternal border (Erb‘s area), and, finally, at the left parasternal area at the lower part of the sternum (tricuspid area) in the supine position. 2) Shift to the diaphragm and return to all these areas. 3) Ask the patient to exhale completely and stop breathing, listen at the apex and aortic areas, and Erb‘s area with the bell and the diaphragm; ask the patient to inhale completely and stop breathing, listen at the pulmonic and tricuspid areas with the bell and the diaphragm. 4) Ask the patient to roll partly onto the left side, listen at the apex with the bell and the diaphragm, then also ask the patient to exhale completely and stop breathing, and again listen with the bell and the diaphragm. 5) Ask the patient to sit up, lean forward and put his/her arms on the head, listen at the aortic and Erb‘s areas only with the diaphragm, also repeat the auscultation at these areas in full held expiration. Physical conditions for all listeners were as in a real-life: all auscultations were performed in an ordinary room of patients.
- Were physicians allowed to listen at a particular site more times until they are happy with the sound quality or were they allowed only to listen once at a certain spot? This is important since we should know what was the auscultation baseline in this study
Seeking to do the study as bosom to real-life clinical practice as possible, physicians were allowed to listen at a particular site more times until they were happy with the sound quality during one auscultation. We added this explanation to Methods.
- While I do understand that this is a pilot study, limitation of having only two physicians and residents and only 30 patients (despite generating a lot of data points) is limiting conclusion strength and should be emphasized as a major limitation
The limitation of having only 2 physicians and 2 residents as well as only 30 patients (despite generating a lot of data points) we added to Limitations as a major limitation.
- While authors did mention chief reasons of hospitalization of their 30 subjects, mostly being admitted for dyspnea and chest pain, I would advise authors to prepare a Table that would contain all baseline characteristics of variables that were recorded for this patient cohort, as well as disease entities (e.g. heart failure, etc.) – were these patients ambulatory or acute?
All the patients were hospitalized, and they were not ambulatory patients. We prepared Table 2 containing all recorded baseline characteristics of the patients, according to your suggestion.
- Figure 1. Please change the coloring of the columns and make them more distinct. It is difficult to follow what is going on over there and I think our readers would appreciate the more colorful depiction of this figure since it is very difficult to follow it in this current form.
We performed the more colorful depiction of the figure 1 (now it is figure 2) since it was really very difficult to follow it in the previous form.
Round 2
Reviewer 2 Report
All my concerns were addressed effectively and manuscript is much improved now.